# How Compounding Pharmacies Fill Critical Gaps in Pediatric Drug Development Processes: Suggested Regulatory Changes to Meet Future Challenges

**DOI:** 10.3390/children9121885

**Published:** 2022-12-01

**Authors:** Robert B. MacArthur, Lisa D. Ashworth, Keming Zhan, Richard H. Parrish

**Affiliations:** 1Department of Pharmacy Services, Rockefeller University Hospital, New York, NY 10065, USA; 2Department of Pharmacy Services, Children’s Health System of Texas, Dallas, TX 75235, USA; 3Department of Biostatistics, Columbia University, New York, NY 10027, USA; 4Department of Biomedical Sciences, Mercer University School of Medicine, Columbus, GA 31902, USA

**Keywords:** compounding, drug, drug approval, drug shortages, FDAMA, manufactured material, regulation, pediatric, 503A, 503B

## Abstract

Drugs administered to children in the United States fall into two broad categories: (1) those that have followed the US Food and Drug Administration (US-FDA) pediatric drug approval process and are marketed as finished dosage forms with pediatric labeling; and (2) all others, many of which are used “off-label”. The use of most drug products in pediatrics is still off label, often requiring special preparation, packaging, and, in some cases, compounding into preparations. The latter category includes compounded preparations that incorporate either a US-FDA approved finished dosage form (e.g., a sterile solution, sterile powder, nonsterile capsules, oral solution, crushed tablets, etc.), or rely on bulk active pharmaceutical ingredients (APIs). Compounded preparations are prepared for individual patients in 503A pharmacies, or on a larger scale and not just for specific patients, in licensed 503B establishments. Critical gaps in the current drug approval process for finished dosage forms have created a proverbial “Gordian knot” that needs to be untangled thoughtfully to facilitate increased production and approval of vitally needed medications for pediatric patients. This opinion will describe current regulatory processes pertaining to pediatrics-only drug approval in the United States. Additionally, discussed are steps required for a product to acquire pediatric labeling. Gaps in regulatory approval pathways for both manufactured and compounded pediatric drugs will be identified, especially those that complicate and slow development and availability to patients. Finally, suggestions for regulatory modifications that may enhance pediatric product development strategies for both manufacturers and compounders are suggested.

## 1. Introduction and Statement of the Problem

Great progress has been made in the past 25 years to increase the number of US Food and Drug Administration (US-FDA) approved drugs developed specifically for children. Much of this is attributable to regulatory mandates and incentives enacted via the Pediatric Rule in 1994, the Food and Drug Administration Modernization Act (FDAMA) of 1997, the Best Pharmaceuticals for Children Act of 2002 (BPCA), the Pediatric Research Equity Act of 2003 (PREA) [1,2] and the Rare Pediatric Disease Priority Review Voucher Program of 2012 (PRV) [3]. These regulations and acts have increased investment in pediatric drug research and development and have led to both many important pediatric labeling changes for existing products as well as newly approved pediatric drug products. However, significant gaps still exist. Indeed, the vast majority of approved drug products are not designed or developed for pediatric and infant populations [4]. These finished dosage forms often need to be transformed into compounded preparations at 503A pharmacies (hospitals, home care agencies, retail, specialty medication, investigational, and infusion centers) and 503B outsourcing facilities.

Pediatric compounding performed in these establishments maintains a critical role in filling these gaps. These pharmacies and outsourcing facilities tailor medicine that can be administered to a child (newborns, infants, children, and adolescents), and thereby foster improved patient compliance and safe use [5,6,7,8].

The main objective of this paper is to compare these two pathways to identify regulatory ambiguities and challenges, and to propose regulatory improvements to facilitate further development of manufactured products and compounded preparations designed for children.

## 2. US-FDA Drug Approval Pathway

This section describes the US-FDA drug approval process in general terms. More detailed reviews of the overall drug approval process for New Drug Applications (NDA), Biologics License Applications (BLAs), Abbreviated New Drug Applications (ANDAs), and 505(b)(2) applications are available from other sources [7,9,10,11]. (A subset of this pathway encompasses orphan drug development which will not be included in this review [12].)

### 2.1. Phases for US-FDA Approval 

US-FDA approves a new drug and its labeling for public use only after careful review of data from preclinical studies in animals and human clinical trials which permit an assessment of the risks and benefits of the drug product [13]. US-FDA’s direct involvement begins near the time the drug sponsor submits the results from preclinical testing in an investigational new drug (IND) application. The IND also contains a description of the drug’s chemical, physical, and biochemical properties and factors influencing the pharmacokinetics (PK) and pharmacodynamics of the drug. Additionally, included are plans for human clinical trials, and information about the proposed manufacturing or compounding process. For pediatric products a Pediatric Study Plan needs to be submitted by the end of Phase 2. Human trials can start 30 days after initial submission of the IND unless US-FDA issues a Clinical Hold response.

With an active IND in place, drugs then undergo a multistage process of human clinical testing. The first clinical trials, Phase 1 trials, are designed to assess drug safety, drug absorption, distribution, metabolism, and excretion (ADME), and evaluate different dosing levels. Typically, these studies are performed in a small number of healthy adult volunteers, although under specialized circumstances pediatric patient populations may be enrolled [14]. Phase 2 trials gather initial data about potential drug benefits, (e.g., efficacy) along with continued collection of safety data in a larger number of individuals who have the condition or disease the dosage form is intended to treat. These trials may continue to evaluate a range of doses to determine the optimal dose with respect to both efficacy and safety, suitable for study in Phase 3.

Phase 3 trials are typically randomized controlled trials that compare the safety and efficacy of the drug with a placebo or another drug product approved for the proposed indication. These trials may also study different populations and different dosages of the drug; in some cases, the drug may be studied in combination with other approved drugs to see if the combination improves outcomes. Phase 3 studies vary in size, depending on the number of patient available for study, and the magnitude of the effect of interest; some phase 3 studies are small, while others recruit thousands of research participants. Phase 3 trials are intended to provide the primary clinical evidence of the safety and effectiveness of the drug [15].

If clinical trial results are positive and the drug demonstrates suitable safety and efficacy, the sponsor may file a new drug application (NDA) proposing that US-FDA approve a new drug product for marketing in the United States. The NDA is much more extensive than the IND, and it includes the results of all toxicology and clinical studies. Additionally, in the NDA Chemistry, Manufacturing, and Controls section (e.g., CMC), updated information is included about the active ingredient and drug product (final dosage form) physiochemical properties, which are now much better understood than at the time of IND filing, with detailed descriptions of drug product manufacturing, processing, and packaging. In addition, since many investigational study protocols require some type of compounding for oral or sterile drug delivery, storage and applicable compounding instructions are included. Other NDA sections include identification of the investigators conducting the research and proposed labeling.

If US-FDA approves the drug product, further safety monitoring (phase 4 of the process) continues as sponsors are required to submit regular safety updates to US-FDA, summarizing the adverse event reports and complaints received. Additionally, if US-FDA requires additional post-marketing studies as a condition of approval, sponsors must oversee that work to completion, according to the timeline provided by US-FDA. On average, for innovative drug products, the time from IND filing to drug approval, is 9.1 years (range 5 to 20 years) with an average cost of USD $1.3 billion [10,16]. For pediatric drug products, in addition to the highly regulated, time, and resource intensive IND and NDA requirements noted above, US-FDA has issued a number of pediatric-specific drug development guidance documents (Table 1) that must be taken into consideration. Sponsors must carefully review and apply this guidance where indicated to their IND and NDA programs, to prevent delays in the US-FDA review process, due to noncompliance with guidance document requirements. (https://www.fda.gov/science-research/science-and-research-special-topics/pediatrics. Accessed on 29 November 2022).

The lack of pediatric drug development efforts by the pharmaceutical industry was recognized by legislators and regulators going back to at least the 1970s [17]. Over the course of 40 years, a series of incentive programs tailored for the pharmaceutical and venture capital industries were put into place. These incentives began with the Pediatric Labeling Rule, and thereafter advanced and expanded via The Pediatric Rule, BPCA, PREA, and PRV (Table 2).

Collectively, these programs have been quite successful by many measures. For example, the additional six months of marketing exclusivity granted under BPCA has been applied to 280 US-FDA approved drugs. Under the PRV, drugs for rare pediatric diseases move more quickly through all phases of clinical testing and are more likely to be first-in-class, when compared against drugs for adult rare diseases [18,19]. Notably, all these incentive programs require drug products to advance via the drug approval pathways as described above, with long development times and considerable financial investment.

#### 2.1.1. Role of BCPA for Drugs without Active Patents

BPCA provides funding to National Institute for Child Health and Human Development (NICHD) to fund clinical trials that investigate off-patent approved drugs for pediatric use, with an objective to update product labeling with pediatric-specific safety and sometimes efficacy information [20].

#### 2.1.2. Use of Supplemental New Drug Applications

In pediatrics, large clinical trials studying new chemical entities (NCEs) under an initial IND for a pediatric indication are uncommon. Pediatric labeling is more commonly acquired via the supplementary NDA (sNDA) application process for products that are already US-FDA approved. In this case, rather than large clinical trials, studies required by an sNDA may be limited to single-dose PK bridging (adult to pediatric), and collection of safety data in pediatric populations. Use of bridging studies is limited to circumstances where the pharmacology in adults and children is known and expected to be similar.

An Agency response to an sNDA can include, in some cases, a Complete Response Letter (CRL). As an example of a CRL, in the case of the drug palonosetron, the FDA issued a CRL to the sponsor’s Pediatric Study Plan (PSP) requesting that compounding instructions be added to product labeling (NDA 021372 and NDA 022233), which illustrates how specific the formulation issues can be incorporated in product labeling [21]:

“… a pharmacy compounded palonosetron extemporaneous liquid preparation for oral administration must be developed for chemotherapy -induced nausea and vomiting (CINV-MEC) prevention therapy in pediatric patients. The compounded palonosetron formulation must utilize appropriate oral ingredients (e.g., sweeteners, suspending agents, and diluents). You must also provide the following information for the compounded formulation: active ingredients, diluents, and suspending and sweetening agents; detailed step-by-step compounding instructions; packaging and storage requirements; and formulation stability information. You must provide instructions to pharmacists and health care providers for accurately measuring the volume of IV solution or oral formulation for compounding. Compounding must be performed at 3 or more pharmacies to demonstrate adequacy of the compounding method. Analytical data and statistical results on the compounding method must be provided to the Agency for review [22].”

While there is a pediatric indication for palonosetron, it appears the sponsor did not include any extemporaneous preparation instructions in the approved labeling for either the IV or capsule dose forms [23]. In addition, in 2014, a palonosetron combination product was approved and marketed in oral form as Akynzeo^®^ (e.g., netupitant with palonosetron oral capsules). The initial US-FDA approval in 2014 included no pediatric dosing or liquid formulation and package instructions still do not contain compounding information. Based upon the list of capsule ingredients, it could be a real challenge to compound a stable oral liquid [24] for pediatric use. There are 11 patents assigned to the product [24]. Combined, these factors complicate compounding of palonosetron for pediatric use.

## 3. Drug Compounding Pathway

Throughout its history, compounding has been considered a critical part of the practice of pharmacy. This section focuses on current regulations and practice as shaped by the Drug Quality and Security Act (DQSA). More detailed historical and legal reviews are available elsewhere [25,26,27,28].

### Drug Quality and Security Act of 2013

Title I of the Drug Quality and Security Act (DQSA), called the Compounding Quality Act (CQA), organized state and federal oversight of compounding pharmacies into two categories, as 503A pharmacies and 503B outsourcing facilities. 503A pharmacies are more traditional pharmacies that compound preparations in response to an individual prescription and are state licensed. By compounding in an individualized per-patient manner, limiting the amount of compounding performed in advance of prescription receipt, and limiting the volume of prescriptions dispensed across state lines, these pharmacies are exempt from certain provisions of the Food, Drug, and Cosmetic Act. Those provisions include exemptions from the new drug approval requirement (as described above), labeling with adequate directions for use, and current good manufacturing practice (CGMP) requirements. Note the latter two provisions are also part of the new drug approval process. Oversight of 503A pharmacies rests primarily with the state government(s) that have licensed the pharmacy, although US-FDA has issued a guidance that addresses US-FDA inspections of 503A pharmacies [29].

503B outsourcing facilities were created by the CQA and are permitted to compound preparations in bulk without the requirement of the receipt of an individual prescription. They may compound sterile and nonsterile individual prescriptions from bulk substances and also, unlike 503A pharmacies, may distribute compounded drugs to healthcare facilities and practitioners without first receiving a patient-specific prescription [30]. Per the DQSA 503B outsourcing facilities are not required to license as a pharmacy, but they must pay an annual user fee to US-FDA, comply with CGMP requirements, undergo US-FDA inspections according to a risk-based schedule, report patient reported adverse events to US-FDA, and provide US-FDA with certain information about the preparations they compound. Oversight of 503B outsourcing facilities is performed primarily by US-FDA [27,30].

It is important to note the definition of compounding here to fully understand how the process can benefit individual patient populations, including children. The definition of compounding according to the United States Pharmacopeia (USP) is the preparation, mixing, assembling, altering, packaging, and labeling of a drug, drug-delivery device, or device in accordance with a licensed practitioner’s prescription, medication order, or initiative based on the practitioner/patient/pharmacist/compounder relationship in the course of professional practice (USP <795>) [31].

Drug compounding includes the following:Preparation of drug dosage forms for both human and animal patients;Preparation of drugs or devices in anticipation of prescription drug orders based on routine, regularly observed prescribing patterns;Reconstitution or manipulation of commercial products that may require the addition of one or more ingredients that is not in the package instructions. [US-FDA does not consider reconstitution or manipulation according to the package instructions compounding; andPreparing a medication in accordance with the package insert instructions is compounding under the USP definition, while it is not under US-FDA’s definition [32].

## 4. Reasons for Needed Regulatory Change

Despite good progress made in getting new pediatric products approved by US-FDA, 503A pharmacies and 503B outsourcing facilities still must compound on a large scale to fill the unmet medical needs of children. While the magnitude of compounding for children is unknown, at least 6 billion prescriptions were dispensed in the United States in 2021, and roughly one-quarter were for oral dosage forms. Of these, anywhere from 1–3% of these prescriptions were for compounded sterile and nonsterile preparations. Approximately one-quarter of compounded prescriptions were dispensed to patients aged 0–19 years [33]. Therefore, a reasonable estimate of the total number of compounded oral prescriptions dispensed for children annually is between 3.7 million and 11 million prescriptions. One indication of the increasing interest in pediatric drug compounding is the spike in publications on the topic since the enactment of the DQSA (Figure 1). Despite policies which may facilitate conduct of pediatric randomized clinical trials (RCTs), the publishing gap in high-impact general medical journals between adult and children RCTs is widening. While the DQSA opened up the opportunity to perform non-patient specific compounding of preparations without prior FDA approval, the high regulatory barrier to acquiring US-FDA approval for a new pediatric dosage form hinders research and development efforts.

Most of the newly US-FDA approved pediatric dosage forms are not suitable for the very young (including neonates) and most are not labeled for use in children < 6 years of age, an indicator of a shortcoming of current regulations. On the other hand, compounded dosage forms have been shown to enhance patient compliance to medication regimens and are widely and safely used now [8]. Many of these dosage forms can be made by 503As.

## 5. Challenges for Meeting These Requirements 

Despite successes, there are recognized gaps in the US-FDA pediatric drug development initiatives. Regarding Priority Voucher Review programs (PRV), the Government Accounting Office found few studies that examined the PRV programs, and those that did found the programs had little or no effect on drug development overall [34,35,36]. PRV has maintained a steady annual number and rate, and seems to facilitate advancement from Phase I to Phase II clinical trials for rare pediatric-specific diseases, compared to that for rare adult diseases. A similar trend, however, was not found in later stages of development. Analysis suggests that other policies are needed to expand the pipeline for rare pediatric diseases, particularly by stimulating the entry of new therapies developed specifically for children. Similarly, the PRV program was not associated with a change in the rate of new pediatric drugs starting or completing clinical testing.

Concerns have been raised that delays and a high level of noncompliance with US-FDA post marketing study requests [1]. It would appear that, given the above, resources could simply be placed into supporting compounding, rather than the ongoing efforts to suppress compounding and to replace compounded preparations with much more expensive approved drug products that likely are no more safe or effective.

The cost of drug development ranges from USD $92 million to $1.3 billion [36]. Further, related to cost, the largest global pharmaceutical companies have most of the approved drug products that have received pediatric exclusivity not small startup companies or not for profit companies [37].

Other more effective finished dosage forms that have been developed specifically for children and other special patient groups (e.g., minitabs, buccal films, immediately dispersable tabs, delayed release minitabs) can be prepared at 503B outsourcing facilities or 503A pharmacies with proper equipment and methods in place. More effective in this case means dose delivery by mouth is more reliable and caregiver administration is easier. Compounded pediatric preparation formulations can be developed at 503As and Bs that can meet high quality production standards in accordance with USP chapters <795> and <797>. 503A pharmacies are able to compound preparations of complex solutions and suspensions (sterile and non-sterile), lollipops, troches, and buccal preparations. Compounded dosage forms have also been shown to enhance patient compliance or adherence to medication regimens [8].

Regarding post-marketing study requirements often put into place by US-FDA to prompt pharma companies to eventually complete pediatric research commitments, justifiable concerns have been raised about delays and noncompliance with these study requests. For example, overall, 64 percent of new drug and biologic indications deemed relevant to pediatric patients lacked pediatric prescribing information at five years after US-FDA approval [1].

The costs to consumers for the BPCA, PREA, and PRV have been exceptionally high, in the 100′s of millions of dollars per approved drug product. Similarly, the standard level of patent protection (note many 505b2 products have patent protections as well as 505b2 exclusivity) commonly in place for 505b2 approvals (i.e., sNDA) adds another time and cost barrier to development. When the target pediatric population is smaller, it is more difficult to justify the cost of patent and product development. One suggested alternative is that policymakers should consider direct funding of such studies rather than shifting much higher product costs to pediatric patients [18]. Some industry executives have held that the slow rate of US-FDA approvals of pediatric drug products, due to the high barriers required for approval, create a setting where many drug products used in pediatrics will never be US-FDA approved. Fortunately for pediatric patients, preparations can be made available via the compounding route.

There are at least six drug oral products known to the authors with formulation and compounding information contained in the package insert, including oseltamivir (Tamiflu^®^) capsules, rifampin (Rifadin^®^) capsules, secubitril and valsartan (Entresto^®^) tablets, valacyclovir (Valtrex^®^) tablets, and valsartan (Diovan^®^) tablets. US-FDA could encourage other sponsors to include these instructions in the package insert to facilitate proper preparation of alternate oral dosage forms for children and adults that cannot swallow solid oral doses.

In our view, US-FDA is restraining 503B-registered facilities from addressing pediatric dosage forms. Inclusion of APIs on the US-FDA Drug Shortage List is one requirement before an approved drug can be compounded. This is a major constraint on 503Bs that prevents them from developing dosage forms for pediatric use. Table 3 illustrates several recent examples of approvals via the NDA pathway with associated patents and post-marketing commitments from the sponsor. None of these products were NCEs. Similarly, and more recently, US-FDA Summary Review documents include comments that some newly approved dosage forms provide alternatives to the use of compounded dosage forms (even if not approved for pediatric use). That is, one reason for approval was to replace, with approved products, those that are otherwise compounded [38,39]. Finally, while no cost per dose comparisons are readily available, GMP manufactured approved drug products could be expected to cost more per dose than a compounded preparation.

## 6. Suggestions for Regulatory Change

To begin addressing these gaps in approval pathways, the following suggestions are offered to facilitate and improve the availability of both manufactured products and compounded preparations.


US-FDA could consider allocating additional resources into properly designed surveillance tools that can begin to measure the depth (number of prescriptions) and breadth (number of different dosage forms dispensed) of compounded preparations to better understand the extent to which pediatric patients rely on them. This surveillance system will generate data that can be used to better understand the compounding industry and guide responsive policy. US-FDA’s Compounding Incidents Program that reviews MedWatch adverse event entries for compounded drugs is a first step towards better compounding surveillance in general [40].US-FDA could consider either (a) the creation of a pharmacy compounding grant program for 503A pharmacies and 503B outsourcing facilities in addition to the current grants directed to academic institutions that are involved in compounding practice or (b) foster partnerships between pediatric academic medical centers and registered 503A pharmacies and 503B outsourcing facilities to test and validate evidence-based formulations designed specifically for children.A simplified, safe, and reliable pathway for the authorization of compounded preparations for children, and other special populations, could be developed in conjunction with USP. This can include well defined active ingredient (API) and excipient specifications, along with methods of preparation and finished dosage form tests. Additionally, this pathway can include PK studies (bioequivalence studies, population PK and modeling, among others) to bridge pediatric findings to adult populations, as is currently allowed for many sNDAs.US-FDA could consider facilitating compounded pediatric preparations by not including frequently compounded APIs on the Bulk Substances Used in Compounding list [27]. The Bulks List should be used to incentivize safe compounding practice.US-FDA could encourage the inclusion of alternate dosage form compounding instructions into the labeling of products that are commonly used in pediatrics. The palonosetron CRL, provided previously, is an example. This is especially important when an approved dosage form cannot be used in children, yet the active ingredient is commonly used in pediatrics, and the stability of compounded preparations limited.ANDA filers (e.g., generic drug companies) could be permitted to augment labeling with compounding instructions. This would simply require additional in vitro stability studies of compounded formulations, for example a crushed tablet in a well characterized pharmaceutical vehicle. These studies are relatively simple for the generic manufacturers to perform and could add additional value and labeled uses for their specific product. US-FDA could facilitate dissemination of alternate dosage form stability into the product’s labeling when it becomes known, especially when a product is unstable in aqueous or non-aqueous vehicles.


## 7. Conclusions

Numerous examples exist which illustrate the progress made in pediatric dosage form development. Despite this progress, regulatory challenges continue to stifle development of both US-FDA-approved products and scientifically derived compounded preparations. Notably, the majority of approved products used in pediatrics are used off-label and, in many cases, require compounding prior to dispensing. Specific examples include neuropsychiatric drugs, where development of new molecular entities (NME) for pediatric indications is often an afterthought to the pharmaceutical industry [41]. As another example, while dasatinib bioequivalence studies were conducted in adults using 50 mg dispersable tablets and a 10 mg/mL suspension, the innovator company marketed only solid oral tablets rather than a liquid formulation suitable for pediatric use. Furthermore, the product labeling advises against crushing the tablets even though the official labeling for dasatinib includes an indication for children aged 1 year and older [42]. Similarly, even recently marketed pediatric dosage forms (e.g., oral liquids, suspensions, capsules, sprinkles, and others) may not be appropriate for all pediatric patients due to formulation with inappropriate excipients and are not labeled for all age groups or pediatric populations [43].

In the absence of state regulations or professional or USP educational processes, US-FDA should expand existing surveillance systems to better understand pediatric compounded preparation needs and use. Further, the agency should resource and incentivize registered compounding pharmacies and outsourcing facilities to develop stable, reproducible, and validated preparations, perhaps in partnership with pediatric academic research centers. Finally, a pathway to authorize compounded preparations should be designed, with incentives for 503A, 503B and ANDA filers, that would foster the use of bulk substances as well as permit labeling changes to promote safe compounding practices for pediatric dosage forms.

## Figures and Tables

**Figure 1 children-09-01885-f001:**
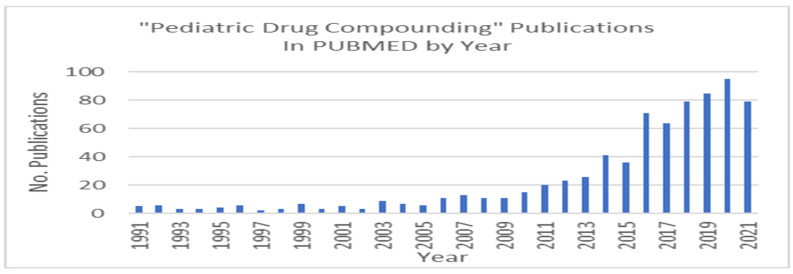
Number of Publications in PUBMED on the topic of pediatric drug compounding, 1991–2021.

**Table 1 children-09-01885-t001:** US-FDA Pediatric Guidance Documents, 2022.

•E11 Clinical Investigation of Medicinal Products in the Pediatric Population
•How to Comply with the Pediatric Research Equity Act
•Nonclinical Safety Evaluation of Pediatric Drug Products
•Over-the-Counter Pediatric Oral Liquid Drug Products Containing Acetaminophen
•Pediatric Gastroesophageal Reflux Disease: Developing Drugs for Treatment Guidance for Industry
•Pediatric Rare Diseases—A Collaborative Approach for Drug Development Using Gaucher Disease as a Model; Draft Guidance for Industry
•Clarification of Orphan Designation of Drugs and Biologics for Pediatric Subpopulations of Common Diseases: Guidance for Industry
•E11(R1) Addendum: Clinical Investigation of Medicinal Products in the Pediatric Population
•Atopic Dermatitis: Timing of Pediatric Studies During Development of Systemic Drugs
•Pediatric HIV Infection: Drug Development for Treatment
•Pediatric Information Incorporated Into Human Prescription Drug and Biological Products Labeling Good Review Practice
•Rare Pediatric Disease Priority Review Vouchers: Draft Guidance for Industry
•Drugs for Treatment of Partial Onset Seizures: Full Extrapolation of Efficacy from Adults to Pediatric Patients 2 Years of Age and Older Guidance for Industry
•Cancer Clinical Trial Eligibility Criteria: Minimum Age Considerations for Inclusion of Pediatric Patients
•Pediatric Study Plans: Content of and Process for Submitting Initial Pediatric Study Plans and Amended Initial Pediatric Study Plans
•S11 Nonclinical Safety Testing In Support of Development of Pediatric Pharmaceuticals: International Council for Harmonization; Draft Guidance for Industry
•FDARA Implementation Guidance for Pediatric Studies of Molecularly Targeted Oncology Drugs: Amendments to Sec. 505B of the FD&C Act: Guidance for Industry
•Development of Anti-Infective Drug Products for the Pediatric Population: Guidance for Industry
•E11A Pediatric Extrapolation
•General Clinical Pharmacology Considerations for Pediatric Studies of Drugs, Including Biological Products
•Ethical Considerations for Clinical Investigations of Medical Products Involving Children: Draft Guidance for Industry, Sponsors, and IRBs

**Table 2 children-09-01885-t002:** Policies promoting evidence-based pediatric drug research and approvals for patented medications [1,17,18,19].

Policy	Description
Pediatric Rule (1994)	- Permitted additions to pediatric labeling based upon extrapolation of efficacy in adult populations together with additional pediatric PK, pharmacodynamic, and safety studies. - Requires that disease and drug response in adults is known to be similar in children. - Compliance is voluntary.
Pediatric Rule in The Food and Drug Administration Modernization Act (1997) (not currently in effect)	- Required sponsors of a new drug to submit, prior to approval, safety and effectiveness information in relevant pediatric population(s) for the claimed indications. However, submission of the pediatric data could be deferred if certain criteria applied. - US-FDA developed a list of drugs where additional of pediatric information would be beneficial. - Sponsors received a Written Request (WR) for pediatric studies along with a required time to completion. - Sponsors that completed the WR requirement received an additional 6 months of marketing exclusivity.
Best Pharmaceuticals for Children Act (2002)	- Provides a financial incentive to companies to voluntarily conduct pediatric studies. - Incentive extends market exclusivity for 6 months after patents expire for pediatrics. - The exclusivity is granted for an active pharmaceutical ingredient and applies to all US-FDA approved drug products containing the ingredient. - US-FDA requests may apply to both approved and unapproved indications. - BPCA is evaluated for renewal every 5 years.
Pediatric Research Equity Act (PREA) (2003)	- Requires sponsors to perform studies involving children to assess the safety and effectiveness of a drug or biologic product for the claimed indications. - Requires studies only for the indication(s) under review in adults. - Applies to applications and supplements for a new active ingredient, new indication, new dosage form, new dosing regimen, and new route(s) of administration. - US-FDA may issue a waiver or deferral for some or all studies involving children. Orphan indications are exempt. - Results guide labeling. - Applies to drugs and biologics.
Rare Pediatric Disease Priority Review Voucher Program (2012)	- Permits US-FDA to issue priority review vouchers to sponsors who receive approval for a product to treat rare diseases in children. - Voucher can be sold and then redeemed to receive a 6-month priority review of a subsequent marketing application for a different product, including products intended only for adult use (e.g., larger patient populations).
Federal Food, Drug, and Cosmetic Act 5059b0(2)	- Other post-approval market exclusivities available after patents expire 5-year exclusivity for new chemical entities; 3-year exclusivity for new clinical investigations; 7-year exclusivity for orphan drugs; and 5-year exclusivity extension under Generic Antibiotic Incentives New (GAIN).

**Table 3 children-09-01885-t003:** Examples of Patents and Post Marketing Commitments for Recent US-FDA Approved Pediatric Dosage Forms via the NDA Pathway https://www.accessdata.fda.gov/scripts/cder/ob/index.cfm. Accessed on 29 November 2022.

Trade and Generic Name	Dosage Form	No. of Patents *	Latest Patent Date	Post Marketing Commitment (Required under PREA) (https://www.accessdata.fda.gov/scripts/cder/pmc/index.cfm)
Carospir^®^ (spironolactone)	Suspension	5	28 October 2036	PMR 3256-1: Conduct a single-dose PK study in pediatric patients 0 to <17 years of age with edematous conditions. PMR 3256-2: Conduct a multiple- dose PK, pharmacodynamics, and safety study in pediatric patients 0 to <17 years of age with edematous conditions.
Kapspargo Sprinkle^®^ (metoprolol succinate)	Capsule, ER ^	2	9 July 2035	PMR 3638-1: Conduct a randomized, dose-ranging, double-blind, placebo-controlled, parallel group, multi-center clinical study with an open-label 52-week safety extension to evaluate efficacy, safety, tolerability and pharmacokinetics of metoprolol succinate extended release (ER) oral dosage form in hypertensive pediatric subjects from birth to less than 6 years of age
Qbrelis^®^ (lisinopril solution)	Solution	8	6 November 2035	PMR 3099-1: An efficacy, safety and dose-finding study of Qbrelis in hypertensive pediatric patients two years to less than six years of age
Katerzia^®^ (amlodipine benzoate)	Suspension	5	11 April 2039	PMR 3640-1: Conduct non-clinical toxicity studies in juvenile rats to evaluate developmental toxicity to include assessment of the effects of amlodipine benzoate suspension on reproductive and learning development to support dosing in humans down to birth. PMR 3640-2: Conduct a dose-ranging, safety, tolerability, and efficacy study with amlodipine benzoate oral suspension in hypertensive pediatric patients age birth to less than 6 years of age.
Norliqva^®^ (amlodipine besylate)	Solution	1	24 February 2041	PMR 4239-1: Conduct a dose-ranging juvenile toxicology and toxicokinetic study to support the definitive toxicology study, and conduct a toxicity study in juvenile rats to evaluate developmental toxicity, including potential effects on reproductive development and learning. PMR 4239-2: Conduct an open-label, randomized, single oral dose, two-treatment, two-period, two-sequence crossover bioequivalence and bioavailability study of amlodipine (ethanol-containing) oral solution versus amlodipine (ethanol-free) oral solution in healthy adults under fasting conditions. PMR 4239-3: Conduct a dose-ranging, safety, tolerability, and efficacy study of amlodipine besylate oral solution for the treatment of hypertension in pediatric patients, birth to <6 years of age.

* As reported in the US-FDA Orange Book, October 2022; ^ Extended Release.

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
