# Peer review of "How Compounding Pharmacies Fill Critical Gaps in Pediatric Drug Development Processes: Suggested Regulatory Changes to Meet Future Challenges"

_children, 2022, doi:10.3390/children9121885_

Round 1

Reviewer 1 Report

Dear Authors,

It was a pleasure to read your opinion paper entitled “How compounding pharmacies fill critical gaps in pediatric drug development processes: suggested regulatory changes to meet future challenges”.

This is a very interesting opinion on the regulations that may improve the availability of drugs for larger groups of pediatric populations.

The text is of very good quality. I support its publication after only minor corrections:

1.  There are numerous abbreviations throughout the text. All of them are explained when they appear in the text for the first time. However, sometimes they are used multiple times, and it is not easy to go through the text searching for explanations of the abbreviations.
My suggestion is to prepare the abbreviations list at the end of the article or somewhere near its beginning.

2. Table 1. – my suggestion is to align all the content of this table to the left. It will probably be more readable than now.

3. Table 1., line 7-8: there is a phrase “;Draft Guidance for industry” repeated two times. It should be removed.

4. Page 5, line 157 – there is a double space.

5. Page 9, line 346 – in my opinion, some reference should be added to this paragraph.

With best regards,

Author Response

Reviewer 1:

It was a pleasure to read your opinion paper entitled “How compounding pharmacies fill critical gaps in pediatric drug development processes: suggested regulatory changes to meet future challenges”.

This is a very interesting opinion on the regulations that may improve the availability of drugs for larger groups of pediatric populations.

Authors’ response: thank you very much for your kind comment.

The text is of very good quality. I support its publication after only minor corrections:

  1. There are numerous abbreviations throughout the text. All of them are explained when they appear in the text for the first time. However, sometimes they are used multiple times, and it is not easy to go through the text searching for explanations of the abbreviations. My suggestion is to prepare the abbreviations list at the end of the article or somewhere near its beginning.

Authors’ response: We thank the reviewer for this important comment. We have added an abbreviation glossary at the end of the manuscript.

  1. Table 1. – my suggestion is to align all the content of this table to the left. It will probably be more readable than now.

Authors’ response: We thank the reviewer for this suggestion to make the table more readable. We have changed the table layout accordingly.

  1. Table 1., line 7-8: there is a phrase “;Draft Guidance for industry” repeated two times. It should be removed.

Authors’ response: We appreciate this comment and have removed the duplicate phrase.

  1. Page 5, line 157 – there is a double space.

Authors’ response: we thank the reviewer for detailed review and have removed the space.

  1. Page 9, line 346 – in my opinion, some reference should be added to this paragraph.

Authors’ response: we thank the reviewer for this insightful comment, and have added two citations where compounding was mentioned in the CDER application.

Reviewer 2 Report

1. The introduction is too long and the conclusion is too short. The thesis should be on the change rather than on the detailed explanation how the system works. 

2. The first section is on "Introduction and Statement of the Problem." However, I don't think the problem is clearly stated. 

3. Line 179 not clear. 

Line 194 should be “it is.”  

Author Response

Reviewer 2:

  1. The introduction is too long and the conclusion is too short. The thesis should be on the change rather than on the detailed explanation how the system works.

Authors’ response: we thank the reviewer for this observation. We have shortened the introduction, clarified the main purpose of the paper, and expanded the conclusion.

  1. The first section is on "Introduction and Statement of the Problem." However, I don't think the problem is clearly stated.

Authors’ response: we appreciate the reviewer’s comment, and have clarified the main purpose of the paper at line 66.

  1. Line 179 not clear.

Authors’ response: we thank the reviewer for this comment. However, this is a direct quote from a letter, not something that the authors formulated.

Line 194 should be “it is.” 

Authors’ response: we have entered a space between it and is.

Reviewer 3 Report

Thank you for the opportunity to read and review this manuscript. I have included my comments and feedback below for consideration.

Line 16 - is "finished" the appropriate terminology here?

Line 17- may consider rewording

Line 46 - consider removing "most simply"

Lines 51 and 55 - dasatinib is misspelled

Lines 51 - 59 - would consider removing this section. In the introduction, this appears to be calling out a specific drug and seems incongruous with the general reference made before

Line 60 - consider adding a transition here 

Line 66 - the main objective of "this" paper

Lines 80 - 89 - consider using a reference for this section

Lines 109 - 120 - consider using a reference for this section

Line 134 - unsure of reason for footnote indication?

Line 143 - consider spelling out "six" vs "6"

Table 2 - under Best Pharmaceuticals for Children Act, last bullet - what is the "it" being referred to?

Line 156 - appears to need comma after "In pediatrics"

Lines 186 - 194 - I would suggest removal of this section. This appears targeted and possibly libelous. It also reads as incongruous with the tone of the paper. 

Section 4 - this section seems possibly out of place. May want to consider this as introductory material and move to the top of the paper.

Line 253 - suggest spelling out "million" 

Figure 1 - despite it having the Pubmed notation, does this need a reference?

Lines 289 - 292 - suggest removing repeat citations if not necessary

Line 314 - Suggest rewording the sentence that begins with "As target pediatric population..." for clarity

Line 317 - need a space between sentences

Lines 322 to 327 - does this require a reference or is this based on observation? 

Line 328 - is a reference required here or is this an opinion?

Line 329 - Suggest rewording this sentence for clarity (unsure of meaning in current form)

Lines 335 - 340 -I would suggest removal of this section. This appears targeted and possibly libelous. It also reads as incongruous with the tone of the paper. 

Section 6 - "Suggestions for Regulatory Change" - would consider the use of softer wording; i.e. change "should" to "consider," etc. to avoid pointed and possibly litigious language or take out the designated agency. 

Please note: I am not a lawyer and am attempting to assess this manuscript to the best of my abilities as a peer-reviewer.

Author Response

Thank you for the opportunity to read and review this manuscript. I have included my comments and feedback below for consideration.

Line 16 - is "finished" the appropriate terminology here?

Authors’ response: we thank the reviewer for this observation. However, the term “finished” is codified in FDC Act at 21 USC § 379j-41(7). “(7) The term “finished dosage form” means— (A) a drug product in the form in which it will be administered to a patient, such as a tablet, capsule, solution, or topical application; (B) a drug product in a form in which reconstitution is necessary prior to administration to a patient, such as oral suspensions or lyophilized powders; or (C) any combination of an active pharmaceutical ingredient with another component of a drug product for purposes of production of a drug product described in subparagraph (A) or (B).” We believe that we have used the term correctly.

Line 17- may consider rewording

Authors’ response: we appreciate the reviewer’s comment, and have changed the sentence to “The use of most drug products in pediatrics is still off label, often requiring special preparation, packaging, and in some cases, compounding into preparations.”

Line 46 - consider removing "most simply"

Authors’ reponse: we thank the reviewer for this suggestion, and have changed ‘most simply’ to ‘indeed’.

Lines 51 and 55 - dasatinib is misspelled

Authors’ response: we thank the reviewer for pointing out these misspellings, and have corrected them.

Lines 51 - 59 - would consider removing this section. In the introduction, this appears to be calling out a specific drug and seems incongruous with the general reference made before

Authors’ response: we are grateful for the reviewer’s observation, and have moved the example to the conclusion.

Line 60 - consider adding a transition here

Authors’ response: we thank the reviewer for this suggestion. We have inserted a better transition from the background to the paper’s purpose.

Line 66 - the main objective of "this" paper

Authors’ response: we note this comment, and have changed the word.

Lines 80 - 89 - consider using a reference for this section

Authors’ response: we thank the reviewer for this suggestion, and have added FDA website as a citation for the process.

Lines 109 - 120 - consider using a reference for this section

Authors’ response: reference 15 is attached.

Line 134 - unsure of reason for footnote indication?

Authors’ response: we have removed the footnote.

Line 143 - consider spelling out "six" vs "6"

Authors’ response: we have spelled out 6.

Table 2 - under Best Pharmaceuticals for Children Act, last bullet - what is the "it" being referred to?

Authors’ response: we added ‘BPCA’ instead of it for clarity.

Line 156 - appears to need comma after "In pediatrics"

Authors’ response: we have added a comma after pediatrics.

Lines 186 - 194 - I would suggest removal of this section. This appears targeted and possibly libelous. It also reads as incongruous with the tone of the paper.

Authors’ response: we appreciate the reviewer’s intension, and have omitted those lines.

Section 4 - this section seems possibly out of place. May want to consider this as introductory material and move to the top of the paper.

Authors’ response: we thank the reviewer for this comment. We believe that the two systems of approval and authorization need to be presented first, and have decided to leave as is.

Line 253 - suggest spelling out "million"

Authors’ response: we have spelled out million.

Figure 1 - despite it having the Pubmed notation, does this need a reference?

Authors’ response: we appreciate the reviewer’s insight. We have never seen any manuscript cite PubMed when conducting a search.

Lines 289 - 292 - suggest removing repeat citations if not necessary

Authors’ response: we have removed the duplicate citations.

Line 314 - Suggest rewording the sentence that begins with "As target pediatric population..." for clarity

Authors’ response: we appreciate the reviewer’s observation, and have changed the sentence to “When the target pediatric population is smaller, it is more difficult to justify the cost of patent and product development.”

Line 317 - need a space between sentences

Authors’ response: we inserted a space.

Lines 322 to 327 - does this require a reference or is this based on observation?

Authors’ response: we added the phrase “known to the authors” to read “There are at least six drug oral products known to the authors with formulation and compounding information contained in the package insert,”

Line 328 - is a reference required here or is this an opinion?

Authors’ response: we appreciate the reviewer’s thought, and have added “In our view”

Line 329 - Suggest rewording this sentence for clarity (unsure of meaning in current form)

Authors’ response: we have reworded the sentence.

Lines 335 - 340 -I would suggest removal of this section. This appears targeted and possibly libelous. It also reads as incongruous with the tone of the paper.

Authors’ response: we thank the reviewer for this suggestion. We have removed specific reference to sildenafil.

Section 6 - "Suggestions for Regulatory Change" - would consider the use of softer wording; i.e. change "should" to "consider," etc. to avoid pointed and possibly litigious language or take out the designated agency.

Authors’ response: we appreciate the reviewer’s insights, and have changed the language to “consider” and “could.” We believe it is important to name the agency, since lawmakers and regulators need to be aware of this.

Please note: I am not a lawyer and am attempting to assess this manuscript to the best of my abilities as a peer-reviewer.

Authors’ response: we have appreciated all of the reviewer’s comments and suggestions.

Round 2

Reviewer 2 Report

The authors have done a good research on this topic. The changes made are good. I still think more work can be done in improving the latter part of the essay. In essence, what makes the essay original is not describing or presenting what has existed and but addressing and solving the problems. 

Reviewer 3 Report

Thank you for your kind remarks in response to my comments, and for the opportunity for a second review of this paper. I feel that the revisions have enhanced the quality and flow, and I wish you all the best with your publication.